# Influence of Environmental and Productive Factors on the Biodiversity of Lactic Acid Bacteria Population from Sheep Milk

**DOI:** 10.3390/ani10112180

**Published:** 2020-11-22

**Authors:** Álvaro Rafael Quintana, José Manuel Perea, María Llanos Palop, Ana Garzón, Ramón Arias

**Affiliations:** 1Regional Institute of Agrifood and Forestry Research and Development of Castilla-La Mancha (IRIAF), CERSYRA Valdepeñas, 13300 Ciudad Real, Spain; rarias@jccm.es; 2Department of Animal Production, University of Córdoba, 14071 Córdoba, Spain; pa2pemuj@uco.es (J.M.P.); pa1gasia@uco.es (A.G.); 3Department of Analytical Chemistry and Food Technology, Faculty of Environmental Sciences and Biochemistry, University of Castilla-La Mancha, 45071 Toledo, Spain; MariaLlanos.palop@uclm.es

**Keywords:** lactic acid bacteria, LAB, dairy farm environment, ewe’s milk, farming practices

## Abstract

**Simple Summary:**

The dairy sheep sector in Spain is of great importance in the socioeconomic field. For this reason, obtaining quality milk has become a priority objective in the sector. In this context, the environment of dairy farms could affect the microbial communities present in milk, and therefore, the study of lactic acid bacteria (LAB) in this environment could be fundamental for the quality of milk and its dairy products. The objective of this study was to investigate the LAB population present in dairy sheep milk and the possible routes of contamination between the livestock environment and the milk on 12 sheep farms with different livestock practices in Castilla-La Mancha. The results showed that certain agricultural practices favour the presence of LAB in milk in addition to the fact that a significant transference between the livestock environment and bulk tank milk could exist.

**Abstract:**

Milk is a typical and satisfactory medium for the growth of lactic acid bacteria (LAB). These microorganisms are of vital importance in the quality of the milk since they contribute to its preservation and give differential organoleptic properties to the final product. Furthermore, LABs can act as biocontrol agents in the dairy industry by inhibiting the growth of undesirable bacteria present in milk and by improving the quality of dairy products such as cheese. In this context, knowing the transfer routes used by LABs from the livestock environment to the milk is of great importance within the dairy industry. Therefore, the objectives of the present study were to expand the knowledge of the LAB population present in the milk of Manchego ewe by means of DNA sequencing techniques and to evaluate the possible transfers of LAB species based on the management of each dairy farm. Samples of bulk tank milk, air (from the milking parlour and from the livestock housing), animal feed and teat surface (taken from 10 sheep per farm) were collected in 12 traditional livestock farms in Castilla-La Mancha (Spain), where each farm presented differences regarding their farming practices. A mixed-effects model was used to evaluate the effects of livestock practices on the distribution of LAB species. Results showed that the vast majority of species identified in the milk had an isolate that was also found in other matrices, which could indicate a microbial transference via the livestock environment to the milk. In addition, the mixed model showed that the factors that positively influence the LAB count were the low-line milking system and the daily use of acid detergent in cleaning the milking machine.

## 1. Introduction

The dairy sector in Spain is particularly important, both in the agri-food sector and in the country’s social sphere, contributing to the sustenance and development of the rural population. In this sector, 87.7% of the total milk produced comes from cows, 6.3% comes from sheep and 5.9% comes from goats [1]. Despite the fact that sheep milk represents a low percentage in our country, the contribution of Spain to the total European production of this type of milk is relevant. Currently, Spain is the third largest producer of sheep milk in Europe [2], representing 18% of total production. Castilla-La Mancha is an autonomous community in Spain where dairy sheep farming, with the Manchega breed being predominant, has traditionally been defined as a mixed cereal-sheep system, generally with semi-extensive family farms that evolve towards greater development and for which the main objective is the production of milk for the manufacture of cheese with the protected designation of origin “Queso Manchego” [3,4]. A detailed description of the sheep milk production system in this region can be found in Rivas et al. [5]. The most recently published data indicates that, in 2018, there were 547,737 Manchega sheep distributed throughout 665 farms [6].

In that sense, one of the main objectives of the dairy sector is to obtain high-quality milk, thus protecting food safety and providing the industry with milk with the best characteristics for processing [7]. Factors related to the microbiological characteristics in the environment of dairy farms as well as those with an influence on the quality of bulk tank milk could affect the microbial communities in milk [8,9], playing a decisive role in the quality of the milk and that of its dairy products. The routes of transmission of microorganisms from the air to contaminate the milk remain a target in research on dairy farms. The routes of contamination used by microorganisms to alter the microbiota of milk have not been deeply studied in cattle [10], and little is known about small ruminants [11]. In addition, the importance of different factors that affect bulk tank milk quality and environmental quality of dairy farms must be taken into account when considering the existing microbiota in the environment as a tool for improving the quality of milk and its products [12,13]. Adequate environmental hygienic-sanitary conditions could positively influence the microbiological quality of milk by reducing the populations of undesirable bacteria present in the livestock. This is mainly due to the air present in the environment, which is a hostile environment as a habitat for the growth of microorganisms, but it is an important vehicle that contributes to its dispersion in which all kinds of microorganisms and their metabolites can be found [14]. Because of that, it is important to focus on the air quality of livestock facilities as this may be one of the main routes of milk contamination on dairy farms.

The microbiological quality of milk has relevant influence on processed dairy products, particularly in the cheese making process [15]. The microbiota frequently found in milk is made up of a wide variety of bacteria, including lactic acid bacteria (LAB) and other undesirable ones that can pose a health risk, and fungi, capable of interacting, playing a decisive role during the manufacture and maturation of cheese [16,17].

LABs are responsible for the initial fermentation of milk lactose-producing lactic acid, leading to a drop of around 5 in pH which prevents the growth of pathogens in cheese maturation, contributing to food safety [18]. In addition, they carry out another essential function which is the formation of amino acid and ammoniacal nitrogen through the degradation of peptides and the subsequent metabolism of amino acids, playing an important role in degradation of the lactic protein during the manufacturing process. LABs, in addition to contributing to food preservation, improve their sensory characteristics such as taste, smell and texture and increase their nutritional quality [19]. It has even been claimed that they can exert some antimicrobial activity by inhibiting the growth of undesirable bacteria such as staphylococci [15,20]. Additionally, several LAB strains are potential conjugated linoleic acid (CLA) and gamma-aminobutyric acid (GABA) producers and therefore candidate for starter cultures with the capacity to generate bioactive compounds, offering new possibilities for manufacturing functional dairy products [21,22]. For the above reasons, it is essential to know about the LAB population present in the livestock environment and their influence on milk and derived products.

In this context, the objectives of this work are to characterise the population of the dominant LAB species in Manchega sheep’s milk and to determine the influence of environmental and productive factors on them. The data collected in this study will provide useful and critical information on the distribution of the dominant species of LAB in the dairy sheep farms of the Manchega breed. Therefore, they should serve as a basis for future decisions on the control of these populations in relation to milk quality.

## 2. Material and Methods

### 2.1. Study Design

Twelve herds of this traditional dairy system from Castilla-la Mancha were selected from among those belonging to the National Association of Selected Breeding of Manchego Sheep (AGRAMA), which represents 10% of the AGRAMA Breeding Program farms with different milking and feeding practices. Another criterion applied to the selection of livestock was that they were within a radius of 200 km away from the CERSYRA Lactology Laboratory in Valdepeñas (Ciudad Real) because this would allow for rapid analysis of the samples and would avoid their deterioration. The differences between farms were mainly based on infrastructure, hygienic conditions and type of diet.

The farms were visited 4 times in 2018, during the months of February, May, August and November, and samples of milk; air from the livestock housing; and air from the milking parlour, animal feed and teat surface were taken. During each visit, an air sample was also taken from the room during milking. Another air sample was collected in the livestock housing where the milking sheep rest before milking. During sampling, the orientation and size of the milking parlour and the livestock housing of the milking sheep, the number of openings in each of these buildings, the handling and characteristics of the milking machine, and the environmental conditions were recorded.

Air samples were obtained using the AirPort MD8 sampler (Sartorius Stedim Biotech, Sartorius, Goettingen, Germany), which operates through the filtration and impact method. The sampler was placed at a minimum distance of 1 m above the floor and from the walls or any existing obstacle in the room [23]. In the milking parlour, the sampler was also located near the teat cups. A defined volume of air, 1000 L, was passed through a gelatin membrane filter (Sartorius Stedim Biotech) where airborne bacteria are retained. The filters were then placed directly on the surface of a culture medium suitable for the growth of the LABs in a Petri dish in order to carry out the counts of these bacteria. Two samplings were performed on each occasion, and the gelatin membrane filters were placed in duplicate directly onto Man, Rogosa and Sharpe (MRS) plates and incubated as described later.

In the milking parlour, 10 sheep were randomly selected to take samples from the teat surface (the area that can come in contact with the teat cup) before milking, following the protocol described by Vacheyrou et al. [10]. A sterile wipe moistened with sterile saline solution (NaCl 0.9% *w*/*v*) was used for the surface for each sheep and was placed in a sterile plastic bag after sampling.

The milk sample was taken from the bulk tank after milking, having stirred and homogenised it for 5 min, using two sterile 50-mL containers.

Finally, during each visit to the farm, a sample of the sheep feed was taken directly from the feeders immediately after dispensing and was placed in a sterile plastic bag.

All samples were transferred to a polystyrene box for transfer to the laboratory under refrigerated conditions at 4 °C. A total of 240 samples (20 per farm) were taken, 48 samples (4 per farm) from each of the following matrices: air from the milking parlour, air from the livestock housing, bulk tank milk, teat surface and animal feed.

Additionally, a survey was conducted with the farmers to collect information about their livestock practices, the estimated number of sheep, milk production, the sanitary conditions of the livestock, general maintenance practices and the feed given to the animals.

### 2.2. Sample Handling and Microbial Counts

To carry out the LAB analysis on all samples, the Man, Rogosa and Sharpe (MRS) medium (Oxoid) supplemented with 50 mg/L sodium azide and 100 mg/L cycloheximide (Sigma, St. Louis, MO, USA) was used to avoid growth of acetic bacteria and yeasts, respectively [24]. The plates were incubated at 30 °C for 5 days under anaerobic conditions (Thermo Scientific™, Oxoid AnaeroGen™, Basingstoke, UK).

The sterile wipes used for sampling the teat surfaces were placed in 50 mL of saline and mixed in a Stomacher (Masticator, IUL S.A., Barcelona, Spain) for one minute at 1.400 rpm. Serial dilutions in buffered peptone water (1 g/L) (Panreac, Barcelona, Spain) were made from this suspension and plated on the surface of MRS agar plates in duplicate. Ten grams of the animal feed samples were homogenised in 90 mL of sterile saline in a Stomacher for 1 min, and the procedure was the same as that for the teat surface analysis. Milk samples were diluted in sterile saline and plated in duplicate on the surface of MRS agar plates.

After incubation, colonies present on the plates were counted. LAB counts were normalised by transformation into log_10_, and the results were expressed as the average number of those counted on each of the plates in the units indicated by Wehr and Frank [25]: log_10_ CFU per mL in the case of milk samples, log_10_ CFU per g for animal feed samples, log_10_ CFU per wipe for teat surface samples and log_10_ CFU per 1000 L air for air samples.

A representative number (10%) of colonies grown on all the sample plates were randomly picked and propagated until purification on MRS plates. Pure cultures were stored at −80 °C in MRS broth containing 20% (*v*/*v*) glycerol (Panreac).

### 2.3. RAPD-PCR Analysis and Identification of Isolates

Pure cultures were analysed by Randomly Amplified Polymorphic DNA-Polymerase Chain Reaction (RAPD-PCR) according to the procedure described by Ruiz et al. [26]. Genomic DNA extraction was carried out as described by Rodas et al. [27]. The primer M13 (5′-GAGGGTGGCGGTTCT-3′), purchased from Bonsai Technologies Group (Madrid, Spain), was used. Amplification was carried out using a 2400 Perkin Elmer thermal cycler (Perkin Elmer Co., Waltham, MA, USA). Amplified products were resolved by electrophoresis (50 A for 3 h, without cooling) on 1.5% (*w*/*v*) agarose in 1 × Tris Borate EDTA (TBE, Tris 89 mM; Boric acid 89 mM; EDTA 2 mM) buffer gels, stained with ethidium bromide and photographed with a Kodak DC290 Digital Camera. A 100-bp ladder (Biotools, Madrid, Spain) was used as a DNA molecular weight marker and as a normalization reference. RAPD-PCR gels were visualized by UV trans-illumination at 254 nm and photographed with a KODAK DC290 Zoom Digital Camera. The patterns were normalized and further processed using the GelCompar version 4.0 pattern analysis software package (Applied Maths, Kortrijk, Belgium).

A reproducibility study to determine the minimum percentage of similarity necessary for pattern discrimination was carried out as reported by Seseña et al. [28] on six isolates and with four iterations of the entire procedure beginning with culture inoculation. Each isolate was grown in four separate cultures from which DNAs were extracted and amplified. The amplification products obtained for two replicates of each isolate were run on one gel, and those for the other two replicates were run on another different gel to estimate gel effects. The level of similarity obtained between repeats, when included within the dendrogram for all isolates, established a discrimination threshold below which patterns were considered different.

The similarity of patterns was expressed by the Pearson correlation coefficient (*r*), and clustering was performed by the unweighted pair group method using average linkage (UPGMA; Gel Compar, Comparative Analysis of Electrophoresis Patterns, Version 4.0, Applied Maths/Kortrijk) as described by Vauterin and Vauterin [29].

Ten percent of the isolates included in each RAPD-PCR cluster and those with a unique pattern were analysed by MALDI-TOF mass spectrometry by Probisearch S.L. (Fundación Parque Científico de Madrid, Spain). Identification was defined with 99–100% agreement with species-specific m/z profiles in the database.

### 2.4. LAB Biodiversity Study

To determine the species richness in each livestock herd, the Simpson biodiversity index was calculated, which takes into account both the number of species present and the relative abundance of each of the LABs found. As species richness and uniformity increases, so does diversity. The following equation was used:(1)$$D=1-\left(\frac{\sum n\left(n-1\right)}{N\left(N-1\right)}\right),$$
where D is the Simpson diversity index, n is the total number of organisms of a species and N is the total number of organisms of all species in the same environment. The D value ranges from 0 (no diversity) to 1 (infinite diversity).

In addition, the percentage of biodiversity of the species was calculated as the rate between the number of different RAPD-PCR patterns obtained for the isolates of a species and the total number of isolates of the species, where the higher values indicate greater diversity.

### 2.5. Statistical Analysis

A mixed model (MIXED procedure in SAS version 9.1, SAS Institute Inc., Cary, NC, USA) was used to examine the factors influencing the variation of log_10_ LAB in milk (LAB-M) [9]. The model could be expressed as follows:(2)$$LAB-M=\sum_{i=1}^{n}\beta_{i}f_{i}+\delta ,$$
where LAB-M is the log_10_ of the LAB count in milk, βi is the unknown parameters to be estimated, fi is the explanatory variables and δ is the error term. The model was developed following the methodology described in detail by Quintana et al. [11].

The factors evaluated were the following: season (spring, summer, autumn or winter), hygiene of the milking parlour (HygMP: adequate or not), hygiene of the livestock housing (HygLH: adequate or not), milking parlour orientation (OriMP: north–south, east–west, northeast–southwest, northwest–southeast or other), livestock housing orientation (OriLH: north–south, east–west, northeast–southwest, northwest–southeast or other), ventilation of the livestock housing (VentLH: adequate or not), milking parlour cleaning frequency (CleanMP: twice a day, daily or less frequent), milk filter change frequency (filter: for each milking, daily or every other day), milk pipeline height (milkline: mid-line or low-line), contact of the teat cups with the ground (cluster: yes or no), frequency of use of acid to clean the milking machine (acid: daily, every 2–3 days or less frequent), use of silage in sheep feed (silage: yes or no) and use of grain during milking (grain: yes or no). LAB-A1 and LAB-A2 showed a strongly nonnormal distribution, so it was decided to scale them into a categorical variable with two levels (0 = absence of the microorganism; 1 = presence), while LAB-F and LAB-T were analysed as continuous factors. The conditions used to determine adequate ventilation and adequate hygiene have been described in Quintana et al. [11].

## 3. Results

### 3.1. Microbial Counts and Identification of LAB Isolates

The presence of microorganisms of special technological interest, such as LAB, was detected in all the matrices analysed. The LAB counts were higher in animal feed samples, followed by milk and teat surface samples, and the matrices with the lowest counts were those of the air from the milking parlour and from the livestock house (Table 1), which indicated facilities with a slightly contaminated environment.

A total number of 703 isolates were obtained from MRS plates from all samples. RAPD-PCR with M13 primer displayed 304 different patterns at an *r* value ≥ 86%, obtained in the reproducibility study (data not shown). MALDI-TOF analysis of isolates from these clusters allowed identification of species of the genera *Lactobacillus* (*Lb.*), *Pediococcus* (*P.*), *Streptococcus* (*S.*), *Lactococcus* (*Lc*), *Leuconostoc* (*L.*), *Weissella* (*W.*) and *Enterococcus* (*E.*), with an identification probability of 99.9%. In all samples, at least one species from each of these genera was found, with the exception of the genera *Streptococcus* and *Lactococcus*, for which the species were only found in the milk samples and in the milk and air samples of the milking parlour, respectively.

The predominant genus was *Lactobacillus,* with a percentage of 74% of all identified species, while the genus *Pediococcus* accounted for only 8%. *Lb. pentosus/plantarum/paraplantarum* was the most important species identified in the study, representing a total of 32.4% of all species, followed by *Lb. curvatus*, which represented 25% of the total. Other minority species of the *Lactobacillus* genus were *Lb. casei/paracasei/rhamnosus* (7.5%), *Lb. brevis* (4%), *Lb. fermentum* (3.6%) and *Lb. coryniformis subspp coryniformis* (1.5%). The species of the genus *Pediococcus* found were *P. acidilactici* (4.7%) and *P. pentosaceus* (3.3%).

The *Enterococcus* genus represented 8.8% of all identified species. Within this genus, we found the following species: *E. hirae* (6.7%), *E. faecalis* (0.9%), *E. faecium* (0.7%), *E. casseliflavus* (0.4%) and *E. mundtii* (0.1%).

The *Weissella* genus represented 6.4% of the total with two species identified, *W. confusa* (5.8%) and *W. paramesenteroides* (0.6%), while the *Leuconostoc* genus only represented 1.8% of the total with three identified species: *L. citreum* (1.3%), *L. mesenteroides* (0.4%) and *L. pseudomesentereoides* (0.1%).

Finally, the *Lactococcus* genus represented 0.6% with a species identified only in the milk and air samples from the milking parlour, namely *Lc. Lactis*, while the genus *Streptococcus* with 0.4% is represented by two species, *S. gallolyticus* ssp. *gallolyticus* (0.3%) and *S. infantarius subspp infantarius* (0.1%), only identified in the milk samples.

The species *Lb. pentosus/plantarum/paraplantarum* was the most representative in 4 of the samples analysed, followed by *Lb. curvatus*, and together always represented more than 50% of the total species present in each sample. *Lb. curvatus* was at the top in terms of percentage of *Lb. pentosus/plantarum/paraplantarum* only in the case of the sample of the teat surfaces. The other species presented high variability in terms of percentages in the different samples. Figure 1 shows percentages of the species of each of the genera identified for each of the matrices analysed.

From each of the identified species, between 1 and 46 different isolates were found. In spite of RAPD-PCR analysis carried out in this study being considered insufficient to assign isolates to different genotypes, it is interesting to mention that clusters obtained from the UPGMA analysis of the patterns grouped isolates from different matrices, with a high percent similarity, which might support the idea that cross-contamination occurs.

### 3.2. LAB Biodiversity Study

The biodiversity of species in each livestock, calculated as the Simpson index (D), is shown in Table 2 with values that ranged between 0.71 and 0.85.

The highest species richness was found on dairy farm F7 (0.852) with 12 identified species, followed by farms F9 (0.829), F1 (0.827), F2 (0.819) and F12 (0.804) which presented between 11 and 9 species. The other farms showed a D index ranging from 0.700 to 0.800, with F8 (0.778), F3 (0.775), F10 (0.759), F6 (0.748), F5 (0.729), F11 (0.728) and F4 (0.714) being those that presented a smaller number of identified species. Existence of a predominant species, as occurred with *Lb. curvatus* in F4 farm or *Lb. pentosus/plantarum/paraplantarum* in F11 and F5 farms, could explain the lowest D values obtained for these farms. The D values obtained for each of the matrices are also shown in Table 2. The highest value corresponded to animal feed (0.834), followed by milk (0.826). In contrast, the lowest D value was that corresponding to the air in the milking parlour, which in turn corresponded to the least contaminated matrix (0.734).

The percentage of biodiversity of the species ranged between 4% and 100% (Figure 2). The species with the highest biodiversity values were *S. infantarius subspp infantarius*, *L. pseudomesentereoides* and *E. mundtii*, while those that were in the majority, as is the case of *Lb. pentosus/plantarum/paraplantarum* and *Lb. curvatus*, were the ones that presented the lowest percentages (20% and 4% respectively) due to the fact that some isolates were predominant in most of the samples analysed. In general, those species with five or more isolates presented a percentage of biodiversity lower than 50%.

### 3.3. Factors Related to the Concentration of LAB in Milk

The main characteristics of the farms as collected from the farmers’ responses to the questionnaire are shown in Table 3.

Six categorical variables were significantly associated with the concentration LAB in milk (*p* < 0.05): orientation of the milking parlour (*p* = 0.026), frequency of cleaning of the milking parlour (*p* = 0.001), milk pipeline height (*p* < 0.001), frequency of use of acid for cleaning the milking machine (*p* < 0.001), use of silage (*p* = 0.041) and use of grain during milking (*p* = 0.030). The E–W orientation of the milking parlour, cleaning the milking parlour after each milking, the low milkline configuration of the milking machine and the daily use of acid for cleaning the milking machine tend to increase the concentration of LAB in the milk, while the use of grain during milking and the use of silage tends to decrease the concentration of LAB in the milk. The bivariate association of LAB-M with LAB-F and LAB-T was explored using Pearson’s correlation coefficient, with LAB-T being weakly and positively correlated with LAB-M (r = 0.417, *p* = 0.003).

Two factors were included in the best fitting model for the concentration of lactic microbiota in milk from Manchega sheep dairy farms (Table 4). Significant factors were the milk pipeline height (*p* = 0.001) and the frequency of use of acid for cleaning the milking machine (*p* < 0.001). The concentration of lactic microbiota in milk increases significantly with the low milkline configuration of the milking machine and the daily use of acid for cleaning the milking machine.

The adjusted determination coefficient and the average absolute percentage error reached values of 47.4% and 0.39, respectively. In addition, the variance inflation factor (VIF) of the regression coefficients fluctuated between 1.01 and 1.37, so there is not a multicollinearity problem.

## 4. Discussion

In this study, the characterisation and study of the transmission between environments of the LAB population in dairy sheep farms was carried out. To do so, innovative bacterial identification techniques such as the MALDI-TOF were used, and appropriate statistical models were applied and parameters were analysed to calculate the biodiversity of the identified species. The presence of LAB in the livestock environment had not been studied in depth until now, especially in the case of dairy sheep. This study not only provides new information on the presence of LAB in dairy sheep farms but also establishes an association between the different matrices studied and the analysed farms.

The results obtained from LAB in milk, with mean counts of 78,187 CFU/mL, were practically the same as those obtained in the study carried out by Jiménez [30] in Manchega sheep and were in agreement with those obtained in other studies in bulk tank sheep’s milk [31,32,33]. Likewise, in the set of matrices analysed, the presence of a total of 21 species belonging to the genera *Lactobacillus*, *Lactococcus*, *Streptococcus*, *Enterococcus*, *Pediococcus*, *Leuconostoc* and *Weissella* was revealed, being the most frequently identified in the dairy products, according to the taxonomic criteria of Fox et al. [34]. *Lactobacillus* was the predominant genus in Manchega sheep farming, in accordance with the results of studies carried out on bulk tank milk of other species [35,36]. The genera *Pediococcus*, *Weissella* and *Enterococcus* were important in livestock environments, with the presence of species of the *Enterococcus* genus being notable since they are indicative of milking with hygienic deficiencies [37]. The low percentage of the *Leuconostoc* genus may be a consequence of the nutritional requirements that bacteria of this genus rely on for their growth [38]. Finally, the fact that the *Lactococcus* and *Streptococcus* genera have such a minority representation is surprising and may be due to the fact that the MRS medium used for the counts and/or the incubation conditions used were not the most suitable for their growth and/or due to the PCR biases encountered in such molecular fingerprinting techniques.

The calculation of the Simpson Index allowed for evaluating the richness of the dominant species present in each flock and in each of the studied matrices. The results showed a wide diversity of LAB species in the environment of dairy sheep farms; however, only 5 identified species represented more than 70% of all those found. Nonetheless, it was observed that there is a loss of biodiversity when a species is predominant in the analysed samples, which may be due to the climatic conditions during sampling and to the fact that there are some environments that present unfavourable conditions for some LAB species. In contrast, a high species richness was observed in the milk and feed samples, which may be due to the fact that they are nutrient-rich environments, favoured by the fact that air and animals are natural transport vehicles for LAB dispersion.

Despite the fact that 57% of the isolates identified belonged to the species *Lb. pentosus/plantarum/paraplantarum* and *Lb. curvatus*, their genetic variability turned out to be quite poor due to the presence of a predominant isolate that seems to present greater dispersal and resistance capacity. The same behaviour was observed for *Lb. casei/paracasei/rhamnosus*, *E. hirae* and *W. confusa* with the presence of a dominant isolate in all the matrices studied. These species seem to be common in the environment of dairy sheep farms, since both *Lb. pentosus/plantarum/paraplantarum* and *Lb. casei/paracasei/rhamnosus* have been detected in sheep’s milk [39] and have also been identified in cheeses made from sheep’s milk, with the predominance of *Lb. pentosus/plantarum/paraplantarum* determining the characteristics of artisan cheeses due to their high proteolytic and lipolytic activity [37]. In addition, *Lb. pentosus/plantarum/paraplantarum* and *Lb. casei/paracasei/rhamnosus* are possible CLA-producer isolates and *Lb. brevis* is a possible GABA-producer isolate, being potential candidates for the design of starter cultures [22]. *Lb. curvatus*, *E. hirae* and *W. confusa* have also been identified in various studies of cheeses made from sheep’s milk [40,41]. It should be noted that a coincidence of isolates has been found both in the matrices analysed and in the different dairy sheep farms studied, which could demonstrate the dispersal capacity of these predominant isolates, linked to the studied geographical area. An extensive use of RAPD PCR (more than one primer) in future research could help to trace the contamination routes of the LAB population in the dairy farm environment. Finally, some species identified in this work such as *E. faecalis*, *Lc. lactis*, *S. gallolyticus* ssp. *gallolyticus* and *S. infantarius* ssp. *infantarius* were also reported in sheep bulk tank milk by other authors using 16S rDNA sequencing [42]; thus, a comparison of molecular approaches can be of help to dairy industry in getting a more accurate view of LAB population in sheep milk regarding quality control programs, food safety and new product development. 

The statistical model concluded that the factors influencing the presence of LAB in sheep’s milk were milk pipeline height and frequency of use of acid in cleaning of the milking machine. In our study, the low-line milking system contributes to the presence of LAB in bulk tank milk; however, in another similar study of Manchega sheep carried out by Jiménez [30], no significant differences were observed in the LAB levels between low-level milkline and mid-level milkline milking parlours. This could be due to the differences in the air LAB contamination in function of building or parlour height, as probably LAB values will be higher near the floor (most contaminated environment) than at the highest points. This fact would increase milk contamination in low-level milk pipelines, particularly in milking machines with deficient maintenance or with high air consumption in the milk system, both aspects recently evidenced in Spanish milking facilities [9].

In the same way, the daily use of acid in the cleaning routine of the milking machine favours the presence of LAB in the bulk tank milk because intensive cleaning practices in the flocks probably selects for specific more resistant organisms and, at the same time, deteriorates the integrity of milk liners and tubes, favouring bacterial contamination. Maintaining correct hygiene of the milking facilities improves the eradication of undesirable bacteria [30] and preserves a greater bacterial diversity in the milk, obtaining a more balanced profile between desirable and undesirable bacteria [43]. The frequency of the use of acid in washing of the milking machine has recently been studied as a factor that influences the variation of the total mesophilic count [9] and of yeasts [11] in bulk tank sheep’s milk. Daily use of acidic detergent could cause further deterioration of the liner material and milk tubes, which could facilitate the proliferation and transmission of LAB.

Besides the main effects, the interaction of the significant factors with season was evaluated, finding no significant effects. However, it is possible that significant interactions exist with other factors but, due to the lack of data in some combinations, they could not be analysed. It could be of interest in future researches to expand the experimental design in order to better evaluate the interactions between factors. However, this type of study is highly conditioned by the economic factor.

## 5. Conclusions

In this research, the LAB population and the possible routes used by the LAB species present in livestock environments to reach the bulk tank sheep’s milk have been investigated in relation to the livestock practices carried out in these environments. Positive relationships have been found between the presence of LAB in milk and in different samples from the dairy farms (air in the milking parlour, air in the livestock housing, animal feed and teat surface) due to the fact that common isolates have been found in them. The predominant genus was *Lactobacillus*, with the species *Lb. pentosus/plantarum/paraplantarum* and *Lb. curvatus* representing more than 50% of all the species found. In addition, a relationship has been found between certain farm practices and the presence of LAB in milk, with the population of LAB being increased when the milking system installed on the dairy farm is low line and when acid detergent is used daily in cleaning the milking machine. This investigation should be completed with another in-depth study that would make it possible to know the potential transfers of LAB from other livestock samples to the milk as well as the transfers of these bacterial populations to cheese and the possible role as a biocontrol agent that some species may have, especially those of the genus *Lactobacillus*. Finally, this study shows that the monitoring/surveillance of the air quality of the milking parlour and the livestock housing by means of air samplers can be a very effective tool in quality control programs for sheep’s milk.

## Figures and Tables

**Figure 1 animals-10-02180-f001:**
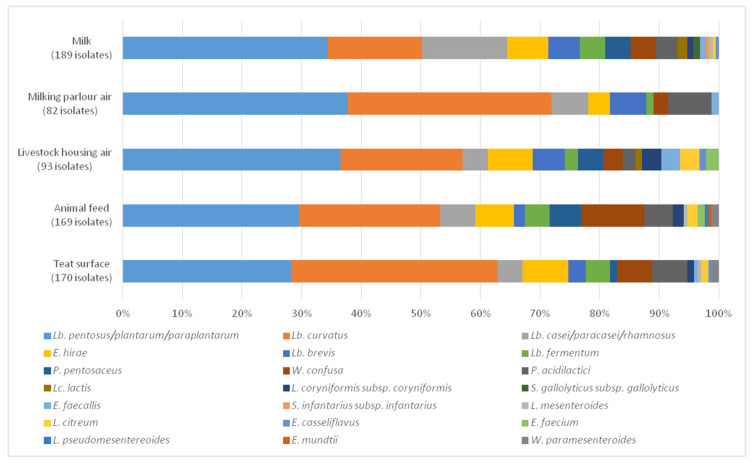
Percentage of each of the species present in the different samples analysed.

**Figure 2 animals-10-02180-f002:**
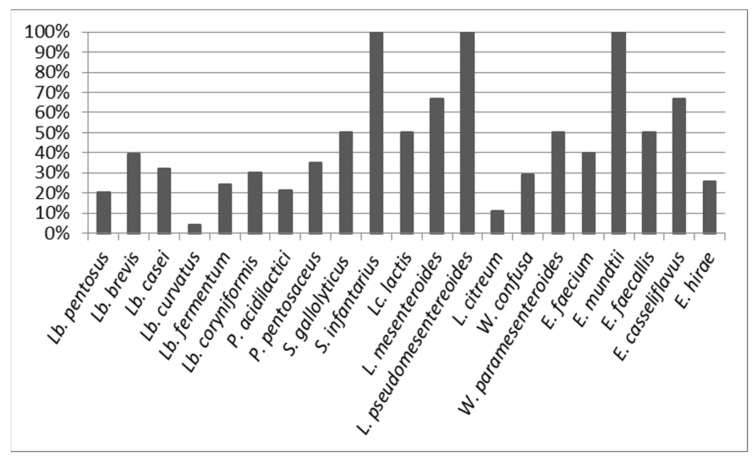
Percentage of biodiversity of the species identified in the analysed samples.

**Table 1 animals-10-02180-t001:** Average counts (log_10_ LAB) in the different samples analysed. Legend: LAB-M = counts of LAB in milk (CFU/mL), LAB-A1 = counts of LAB in the air from the milking parlour (CFU/1000 L), LAB-A2 = counts of LAB in the air from the livestock housing (CFU/1000 L), LAB-F = counts of LAB in the feed (CFU/g) and LAB-T = counts of LAB on the surface of teats (CFU/wipe).

Variable	Mean	Standard Deviation	Coefficient Variation
log_10_ LAB-M	4.21	0.67	15.84
log_10_ LAB-A1	0.64	0.84	132.27
log_10_ LAB-A2	0.84	1.04	123.78
log_10_ LAB-F	4.50	1.36	30.13
log_10_ LAB-T	2.91	0.59	20.19

**Table 2 animals-10-02180-t002:** Simpson index of biodiversity of species on each dairy farm and in each of the matrices: the value ranges from 0 to 1, where 1 represents infinite diversity and 0 represents no diversity.

Simpson’s Index	D Value
Farms	
F1	0.827
F2	0.819
F3	0.775
F4	0.714
F5	0.729
F6	0.748
F7	0.852
F8	0.778
F9	0.829
F10	0.759
F11	0.728
F12	0.804
Matrix	
Milk	0.826
Milking parlour air	0.734
Livestock housing air	0.815
Animal feed	0.834
Teat surface	0.787

**Table 3 animals-10-02180-t003:** Distribution of frequencies obtained by the farmers in the applied questionnaire and bivariate association (ANOVA or *t*-Student test) between the concentration of lactic microbiota in the milk (log_10_ CFU/mL) and the considered categorical factors. Legend: LAB-A1 = presence of LAB in the air from the milking parlour, LAB-A2 = presence of LAB in the air from the livestock housing, Season = season of the year, HygMP = hygiene of the milking parlour, HygLH = hygiene of the livestock housing, OriMP = orientation of the milking parlour, OriLH = orientation of the livestock housing, VentLH = ventilation of the livestock housing, CleanMP = frequency of cleaning of the milking parlour, Filter = frequency of changing of milk filters, Milkline = milk pipeline height, Cluster = possibility of contact between the teat cups and the ground, Acid = frequency of use of acid for cleaning the milking machine, Silage = use of silage and Grain = use of grain during milking.

Variable	Levels	n (%)	Log_10_ LAB-M (Mean + EE)	*p* Value
LAB-A1	No	30 (62.5)	4.14 + 0.12	0.431
	Yes	18 (37.5)	4.31 + 0.16	
LAB-A2	No	28 (58.3)	4.24 + 0.14	0.735
	Yes	20 (41.7)	4.17 + 0.14	
LAB-F	No	0 (0.0)	-	-
	Yes	48 (100.0)	4.21 + 0.67	
LAB-T	No	0 (0.0)	-	-
	Yes	48 (100.0)	4.21 + 0.67	
Season	Spring	12 (25.0)	4.47 + 0.22	0.394
	Summer	12 (25.0)	4.22 + 0.20	
	Autumn	12 (25.0)	4.01 + 0.16	
	Winter	12 (25.0)	4.14 + 0.19	
HygMP	Adequate	35 (72.9)	4.27 + 0.12	0.380
	Not	13 (27.1)	4.07 + 0.16	
HygLH	Adequate	35 (72.9)	4.23 + 0.12	0.787
	Not	13 (27.1)	4.17 + 0.17	
OriMP	N-S	24 (50.0)	4.16 + 0.12	0.026
	E-W	8 (16.7)	4.79 + 0.36	
	NE-SW	8 (16.7)	3.83 + 0.06	
	NW-SE	8 (16.7)	4.18 + 0.19	
OriLH	N-S	12 (25.0)	4.13 + 0.19	0.127
	E-W	16 (33.3)	4.54 + 0.20	
	NE-SW	8 (16.7)	3.83 + 0.06	
	NW-SE	8 (16.7)	4.17 + 0.19	
	Other	4 (8.3)	3.99 + 0.30	
VentLH	Adequate	24 (50.0)	4.29 + 0.15	0.414
	Not	24 (50.0)	4.13 + 0.12	
CleanMP	After each milking	25 (50.0)	4.03 + 0.11	0.001
	Daily	12 (25.0)	4.79 + 0.22	
	Less frequently	12 (25.0)	3.98 + 0.15	
Filter	After each milking	20 (41.7)	4.37 + 0.18	0.365
	Daily	24 (50.0)	4.11 + 0.11	
	Every two days	4 (8.3)	4.04 + 0.10	
Milkline	Mid-level	32 (66.7)	3.99 + 0.09	< 0.001
	Low-level	16 (33.3)	4.65 + 0.18	
Cluster	Yes	24 (50.0)	4.18 + 0.16	0.711
	No	24 (50.0)	4.25 + 0.10	
Acid	Daily	8 (16.7)	5.06 + 0.27	< 0.001
	Each 2–3 days	36 (75.0)	4.06 + 0.09	
	Less frequently	4 (8.3)	3.88 + 0.08	
Silage	Yes	16 (33.3)	4.49 + 0.21	0.041
	No	32 (66.7)	4.07 + 0.09	
Grain	Yes	24 (50.0)	4.42 + 0.16	0.030
	No	24 (50.0)	4.00 + 0.08	

**Table 4 animals-10-02180-t004:** Least squares means of the counts of lactic microbiota in milk (log_10_ CFU/mL) from Manchega dairy sheep farms for factors included in the best fitting mixed model.

Factors	Mean	EE	F Value	*p* Value
Milkline			11.76	0.001
Mid-level	4.18 ^b^	0.11		
Low-level	4.73 ^a^	0.15		
Acid			10.68	<0.001
Daily	5.06 ^a^	0.18		
Each 2–3 days	4.15 ^b^	0.09		
Less frequent	4.15 ^b^	0.26		

Means within factors with different letters differ (*p* < 0.05).

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
