# Peer review of "Influence of Environmental and Productive Factors on the Biodiversity of Lactic Acid Bacteria Population from Sheep Milk"

_animals, 2020, doi:10.3390/ani10112180_

Round 1
Reviewer 1 Report
General comments
In this paper attention is focused on the isolation and main sources influencing the origin and variation of LAB population in native flora of sheep milk. This flora could be successfully used in the dairy industry contributing to food safety and preservation, nutritional characteristics, etc. Therefore, results are relevant for farmers, veterinarians, consumers, and dairy industry, as well as for setting the grounds for further analytical surveillance programs in dairy or cheese industry. Moreover, methodology and statistical analyses are adequate, but the discussion of the results can be substantially improved. Specific comments are below.
Specific comments
L82-92, L220-294: Several LAB strains are potential CLA and GABA producers and therefore candidate for starter cultures with the capacity to generate bioactive compound, offering new possibilities for manufacturing functional dairy products. See, for example: ‘Ogawa et al. (2005). Production of conjugated fatty acids by lactic acid bacteria. J. Biosci. Bioeng., 100: 355-364’, and ‘Renes et al. (2017). Production of conjugated linoleic acid and gamma-aminobutyric acid by autochthonous lactic acid bacteria and detection of the genes involved. J. Funct. Foods, 34: 340-346’. Interest of some LAB strains found in this study can be additionally discussed in this context.
L59-76 and L338-420: Season, breed, milking type, antibiotic dry therapy, etc. were significant effects influencing the variation on different bacterial subpopulations of ewe bulk tank milk as evidenced by other authors, but these results were not mentioned or discussed. Authors should see, for example, ‘de Garnica et al. (2013). Relationship among specific bacterial counts and total bacterial and somatic cell counts and factors influencing their variation in ovine bulk tank milk. J. Dairy Sci. 96: 1021-1029’. In addition, these same authors also study the diversity of acid lactic bacterial species, such as Enterococcus spp., Sreptococcus spp., Lactococcus, spp., etc. in bulk tank milk of Spanish dairy sheep flocks, by comparative 16S rDNA sequence analysis, and an effort should be made for the authors to compare and discuss their results in the present paper. Besides, a comparison of molecular approaches can be of help to dairy industry in getting more accurate view of LAB population in sheep milk regarding quality control programs, food safety and new product development. See, for example, ‘de Garnica et al. (2014). Diversity of gram-positive catalase-negative cocci in sheep bulk tank milk by comparative 16S rDNA sequence analysis. Int. Dairy J. 34: 142-145’.
L187-208: Season is a main source of variation for total and specific bacterial counts, but other factors can also be seasonally affected (e.g., teat surface, teat cup falls, ambient contamination, etc.) or related (e.g., milking parlor orientations). Were some factors/effects analyzed potential confounding variables? Explain, please.
L195-208: LAB contents in faeces and in the floor of milking parlor and sheepfold would have been of interest.
L204-205. I do not understand this factor. Frequency or percentage of teat cup falls would be an interesting variable to be studied, but contact of teat cups with the ground is a non-specific factor, as teat cup falls occurs in all milking flocks. Explain, please.
L216: Logarithmic means and SD should be included in Table 1.
L202, 305, 312, 329, 386: … milk pipeline height or milkline height of the milking machine…instead of…milkline of the milking machine, or type of milking machine line.
L391-395: These are very confuse sentences. Rewrite, please. Mid-level systems are generally dimensioned to a higher working vacuum than low-level systems for carrying milk against gravity. Some comment should be made about the differences in the air LAB contamination in function of building or parlor height, as probably LAB values will be higher near the floor (most contaminated environment) than in highest points. This fact would increase milk contamination in low-level milk pipelines, particularly in milking machines with a deficient maintenance or with high air consumptions in the milk system, both aspects recently evidenced in Spanish milking facilities [9].
L396-397: Intensive cleaning practices in the flocks probably selects for specific more resistant organisms and, at the same time, deteriorates the integrity of milk liners and tubes, favoring bacterial contamination.
L474: Revise, please.
Author Response
Dear Reviewer,
We greatly appreciate your thoughtful comments that helped improve the manuscript. We trust that all of your comments have been addresses accordingly in the revised manuscript.
Thank you very much for your effort.
In the following, we give a point-by-point reply to your comments:
Point 1: L82-92, L220-294: Several LAB strains are potential CLA and GABA producers and therefore candidate for starter cultures with the capacity to generate bioactive compound, offering new possibilities for manufacturing functional dairy products. See, for example: ‘Ogawa et al. (2005). Production of conjugated fatty acids by lactic acid bacteria. J. Biosci. Bioeng., 100: 355-364’, and ‘Renes et al. (2017). Production of conjugated linoleic acid and gamma-aminobutyric acid by autochthonous lactic acid bacteria and detection of the genes involved. J. Funct. Foods, 34: 340-346’. Interest of some LAB strains found in this study can be additionally discussed in this context.
Response 1: We greatly appreciate your thoughtful help with this references. Please see the attachment, lines 90-93 and 374-376.
Point 2: L59-76 and L338-420: Season, breed, milking type, antibiotic dry therapy, etc. were significant effects influencing the variation on different bacterial subpopulations of ewe bulk tank milk as evidenced by other authors, but these results were not mentioned or discussed. Authors should see, for example, ‘de Garnica et al. (2013). Relationship among specific bacterial counts and total bacterial and somatic cell counts and factors influencing their variation in ovine bulk tank milk. J. Dairy Sci. 96: 1021-1029’. In addition, these same authors also study the diversity of acid lactic bacterial species, such as Enterococcus spp., Sreptococcus spp., Lactococcus, spp., etc. in bulk tank milk of Spanish dairy sheep flocks, by comparative 16S rDNA sequence analysis, and an effort should be made for the authors to compare and discuss their results in the present paper. Besides, a comparison of molecular approaches can be of help to dairy industry in getting more accurate view of LAB population in sheep milk regarding quality control programs, food safety and new product development. See, for example, ‘de Garnica et al. (2014). Diversity of gram-positive catalase-negative cocci in sheep bulk tank milk by comparative 16S rDNA sequence analysis. Int. Dairy J. 34: 142-145’.
Response 2: We greatly appreciate your thoughtful help with this references. Please see the attachment, lines 67-69 and 380-385.
Point 3: L187-208: Season is a main source of variation for total and specific bacterial counts, but other factors can also be seasonally affected (e.g., teat surface, teat cup falls, ambient contamination, etc.) or related (e.g., milking parlor orientations). Were some factors/effects analyzed potential confounding variables? Explain, please.
Response 3: We have choosen a main effects model (without interactions) because the sample size does not allow us to evaluate all the interactions between factors. Season was not associated by ANOVA with the dependent variable, neither was it in any model as the main source of variation. However, based on the evaluator's comment, we have analyzed the interaction of season with the factors of the best-fit model (milkline and acid) and they were not significant. According to the evaluator, it is possible that exist interactions between factors that cannot be properly analyzed because we do not have sufficient data in all combinations. However, to carry out this type of study is completely limited by the economic factor, so we believe that identifying main effects is meritorious. We clarify all this in the discussion by including a paragraph, lines 407-412.
Point 4: L195-208: LAB contents in faeces and in the floor of milking parlor and sheepfold would have been of interest.
Response 4: We greatly appreciate your comment. These factors will be taken into account in future researches, in order to improve the knowledge of LAB population in dairy sheep flocks.
Point 5: L204-205. I do not understand this factor. Frequency or percentage of teat cup falls would be an interesting variable to be studied, but contact of teat cups with the ground is a non-specific factor, as teat cup falls occurs in all milking flocks. Explain, please.
Response 5: Again, we appreciate this comment. We classify as "yes" when an expert observer verifies that teat cup falls down when milking young animals, due to improper handling by the milker, deficiencies in breast morphology...
Point 6: L216: Logarithmic means and SD should be included in Table 1.
Response 6: Please see the attachment, Table 1.
Point 7: L202, 305, 312, 329, 386: … milk pipeline height or milkline height of the milking machine…instead of…milkline of the milking machine, or type of milking machine line.
Response 7: We greatly appreciate your comment. Please see the attachment.
Point 8: L391-395: These are very confuse sentences. Rewrite, please. Mid-level systems are generally dimensioned to a higher working vacuum than low-level systems for carrying milk against gravity. Some comment should be made about the differences in the air LAB contamination in function of building or parlor height, as probably LAB values will be higher near the floor (most contaminated environment) than in highest points. This fact would increase milk contamination in low-level milk pipelines, particularly in milking machines with a deficient maintenance or with high air consumptions in the milk system, both aspects recently evidenced in Spanish milking facilities [9].
Response 8: Please see the attachment, lines lines 391-396.
Point 9: L396-397: Intensive cleaning practices in the flocks probably selects for specific more resistant organisms and, at the same time, deteriorates the integrity of milk liners and tubes, favoring bacterial contamination.
Response 9: Please see the attachment, lines 398-400.
Point 10: L474: Revise, please.
Response 10: Please see the attachment, lines 485-487.
Note: some changes have been made from previous edit because we have noticed some errors in the wording.

Reviewer 2 Report
The aim of this study was to monitor lactic acid bacteria in sheep's milk and the pathways of contamination,
If the authors want to trace the contamination routes, they need to do a genotyping study with robust use of RAPD PCR (more than one primer to use) to see if certain bacterial strains could be found at different times and places in the environment.
The RAPD PCR in this work is mentioned in Materials and Methods but the results are presented in two lines (lines 220-221) and appears to be used only to find putative clones from the same samples. For accurate genotyping it is necessary to use more than one primer and to do an extensive discussion on the results of the RAPD PCR, e.g. with genotyping tree. This is why I suggest changing the title and main purpose of the paper to characterize bacterial biodiversity based on the different type of sampling (milk, air) and the different characteristics of the company (type of diet, cleaning system, etc.).
I also suggest i) a better presentation of your data (too many tables and figures to present the same data) ii) a better explanation of the variables examined in this study (the season? the hygiene practices? the diet?) And iii) an extensive improvement of the English language use
Other revisions
material and methods
line 5 please be more detailed: how many km were the sampled farms away from the lab? the time you took to reach them is not a constant variable. And how many farms were excluded with this type of selection?
Lines 7-8 Please make a table with the number of samples collected (240?) sorted by type of sample (air, milk?) infrastructure, sanitation, type of diet and season. Add a column indicating how many farms in the Castilla-la Mancha region have the same infrastructure, hygiene conditions and type of diet to understand how much the number of samples is really representative of the region.
Line9: seasonally means 4 times in a year? Please be more detailed and write the sampling month and not the season
Add a separate chapter to describe the method and equipment used for air, milk and teats sampling and a separate chapter for microbial analysis counts. don't mix sampling and microbial counts
line127 please add the teat area covered with the sterile wipe
Line144 please add stomacher brand model and speed used (not vigorously)
Line 144 serial dilutions in what? Water?
Add chapter for RAPD PCR biotyping (how many primers used? Which ones?) and for species identification
RESULTS
Please analyze the LAB counts by showing the logCFU and not the CFU
Please analyze the data of the counts sorting by type of diet infrastructure and hygienic condition
Figure 1 and Table 2 are a different organization of the same data. Keep only Figure 1 or Table 2 and not both. If you keep Fig 1, add somewhere the number of total isolates in each type of sample, if you choose to keep table 2, please show the number of species isolated from each type of sample and not +/-
move lines 260 to 270 in the discussion section and try to be clearer. The concept is confusing and unclear.
Please make a table instead of Fig 2 and Fig 3 (there are too many figures)
Author Response
Dear Reviewer,
We greatly appreciate your thoughtful comments that helped improve the manuscript. We trust that all of your comments have been addresses accordingly in the revised manuscript.
Thank you very much for your effort.
In the following, we give a point-by-point reply to your comments:
Point 1: This is why I suggest changing the title and main purpose of the paper to characterize bacterial biodiversity based on the different type of sampling (milk, air) and the different characteristics of the company (type of diet, cleaning system, etc.).
I also suggest i) a better presentation of your data (too many tables and figures to present the same data) ii) a better explanation of the variables examined in this study (the season? the hygiene practices? the diet?) And iii) an extensive improvement of the English language use.
Response 1: We greatly appreciate this comments. We have taken this comments into consideration and the manuscript has been checked by a professional English editing service. Please see the attachment.
Point 2: line 5 please be more detailed: how many km were the sampled farms away from the lab? the time you took to reach them is not a constant variable. And how many farms were excluded with this type of selection?
Response 2: We have randomly selected approximately 10% farms of the AGRAMA Breeding Program, within a radius of 200km away (less than 2 hours) depending on the productive characteristics of each farm, which represents 85% of the total AGRAMA farms. Please see the attachment, line 107.
Point 3: Lines 7-8 Please make a table with the number of samples collected (240?) sorted by type of sample (air, milk?) infrastructure, sanitation, type of diet and season. Add a column indicating how many farms in the Castilla-la Mancha region have the same infrastructure, hygiene conditions and type of diet to understand how much the number of samples is really representative of the region.
Response 3: In previous investigations of our research group, the typology of the different farms of the AGRAMA Breeding Program has been evaluated, as can be seen in Jiménez, L. Evaluation of the hygienic-sanitary and technological quality of the milk of the Manchega breed as an instrument for the improvement of the socio-economic and environmental viability of dairy sheep production systems. Doctoral Thesis, University of Córdoba, Spain, 2020. Thanks to this research, we have considered that 10% is representative for the Manchega sheep farms in relation to the productive characteristics of the different farms. Furthermore, we consider that it would be redundant to draw the requested table with table 3 of the article.
Point 4: Line9: seasonally means 4 times in a year? Please be more detailed and write the sampling month and not the season
Response 4: Please see the attachment, lines 111-112.
Point 5: Add a separate chapter to describe the method and equipment used for air, milk and teats sampling and a separate chapter for microbial analysis counts. don't mix sampling and microbial counts.
Response 5: Please see the attachment, line 128.
Point 6: line127 please add the teat area covered with the sterile wipe
Response 6: According with Vacheyrou et al. International Journal of Food Microbiology (2011), we sampled the surface of the teat, the area that can come in contact with the pump. Please see the attachment, line 129.
Point 7: Line144 please add stomacher brand model and speed used (not vigorously)
Line 144 serial dilutions in what? Water?
Response 7: Please see the attachment, lines 148-150.
Point 8: Add chapter for RAPD PCR biotyping (how many primers used? Which ones?) and for species identification
Response 8: The Editor suggest us to consider a shorter text summarizing the main points and referring readers to your previous publication for full details. However, we have rewritten the paragraph. Please see the attachment, lines 162-189.
Point 9: Please analyze the LAB counts by showing the logCFU and not the CFU
Response 9: Please see the attachment, Table 1.
Point 10: Please analyze the data of the counts sorting by type of diet infrastructure and hygienic condition
Response 10: Please see the attachment, Table 3.
Point 11: Figure 1 and Table 2 are a different organization of the same data. Keep only Figure 1 or Table 2 and not both. If you keep Fig 1, add somewhere the number of total isolates in each type of sample, if you choose to keep table 2, please show the number of species isolated from each type of sample and not +/-
Response 11: Figure 1 and Table 2 are not a different organization of the same data. Figure 1 shows percentages of the species identified for each of the matrices analysed, and Table 2 represents the identified species and the relationships found with at least one of the genetically equal strains present in the different samples. However, according to reviewer, we have summarized the manuscript. Please see the attachment, lines 266-275
Point 12: move lines 260 to 270 in the discussion section and try to be clearer. The concept is confusing and unclear.
Response 12: Please see the attachment, lines 266-275.
Point 13: Please make a table instead of Fig 2 and Fig 3 (there are too many figures)
Response 13: Please see the attachment, Table 2.
Note: some changes have been made from previous edit because we have noticed some errors in the wording.

Round 2
Reviewer 2 Report
The paper has for sure been improved but some revisions have not been adequately implemented
As I said in the first review: if the authors wanted to study how the same bacterial strains are present in different sources and times, they should make extensive use of RAPD PCR (more than one primer).
The RAPD PCR in this work does not allow any speculation about genotyping at strain level so please modify the main text according to this. You have to delete/change the main text in many points e.g. lines 268-277; lines 293-299; line 369 (isolates and not strains); lines 372-373
Other revisions
material and methods
Line 107 Please delete less than two hours away
Lines 149-150 change peptone salt solution with peptone water and add the amount of added peptone and brand
Lines 186-187 what do you mean for unique pattern?
Line 189 which database was used? Please report.
Line 200 how do you calculate the number of strains? Again only one primer in RAPD PCR is not enough to have a biotype characterization but only to exclude clones from the same day or sampling source.
Lines 210-211: what do you mean for “adequate” and "not adequate" ?
Lines 268-277 delete it: as said above you could not be sure of the presence of same strains in different samples using only one primer with RAPD PCR: you need to use at least two primer to confirm you have same biotype in different samples. You need more data in order to do a biotyping. With only one primer you have to limit your work to species occurrence and not biotype or strains.
Please in Table 1 keep only the log transformation of the bacterial counts
What do you mean for genetic diversity in Fig 2?
Again in Fig 1, the number of isolates (703) is not as reported in the text (983 at line 235) add somewhere in the Figure the correct number of total isolates in each type of sample
Table 3 change the column Frequency (%) with n (%). Report in the text why your samples are representative of the farms in the Castilla-la Mancha region and how many samples were done for each farm and in total
Author Response
Dear Reviewer,
We greatly appreciate your thoughtful comments that helped improve the manuscript. We trust that all of your comments have been addresses accordingly in the revised manuscript.
Thank you very much for your effort.
In the following, we give a point-by-point reply to your comments:
Point 1: As I said in the first review: if the authors wanted to study how the same bacterial strains are present in different sources and times, they should make extensive use of RAPD PCR (more than one primer).
The RAPD PCR in this work does not allow any speculation about genotyping at strain level so please modify the main text according to this. You have to delete/change the main text in many points e.g. lines 268-277; lines 293-299; line 369 (isolates and not strains); lines 372-373
Response 1: Thanks for your valuable opinion. Although we agree that with two primers the results would have had greater “strength”, the high number of isolates (703) made it impossible to analyze them with two primers. Where possible, we have tried to modify the manuscript following your suggestions. Please see the attachment.
Point 2: Line 107 Please delete less than two hours away.
Response 2: Thanks for your valuable comment. Please see the attachment, line 108.
Point 3: Lines 149-150 change peptone salt solution with peptone water and add the amount of added peptone and brand
Response 3: Thanks for your valuable comment. Please see the attachment, lines 150-151.
Point 4: Lines 186-187 what do you mean for unique pattern?
Response 4: It refers to isolates which were not included at any cluster because their patterns were different to each other.
Point 5: Line 189 which database was used? Please report.
Response 5: Probisearch S.L. sends us the species identification according to its own database.
Point 6: Line 200 how do you calculate the number of strains? Again only one primer in RAPD PCR is not enough to have a biotype characterization but only to exclude clones from the same day or sampling source.
Response 6: The number of strains of a species corresponds to the number of different patterns obtained from RAPD-PCR analysis of the isolates belonging to that species. However, as suggested, we have changed the main text in many points according to this.
Point 7: Lines 210-211: what do you mean for “adequate” and "not adequate" ?
Response 7: The Editor suggest us to consider a shorter text summarizing the main points and referring readers to your previous publication for full details, so the conditions used to determine adequate ventilation and adequate hygiene have been described in Quintana el at. [11]:
The conditions used to determine the inadequate hygiene of the milking parlour were the presence of airborne dust particles, the presence of litter on the floor and the accumulation of waste; the conditions used to determine a inadequate hygiene in the livestock housing were bad litter management, presence of airborne dust particles and leftover in the feeders; and the conditions used to determine inadequate ventilation in the livestock housing were the presence of ammonia odour and the absence of sufficient windows and doors for providing a good ventilation.
Point 8: Lines 268-277 delete it: as said above you could not be sure of the presence of same strains in different samples using only one primer with RAPD PCR: you need to use at least two primer to confirm you have same biotype in different samples. You need more data in order to do a biotyping. With only one primer you have to limit your work to species occurrence and not biotype or strains.
Response 8: We have modified this part according to this comments. Please, see the attachment, lines 272-285.
Point 9: Please in Table 1 keep only the log transformation of the bacterial counts
Response 9: Thanks for your valuable comment. Please see the attachment, Table 1.
Point 10: What do you mean for genetic diversity in Fig 2?
Response 10: Data represented in this figure correspond to percentages of biodiversity of the species. Tittle in the figure has been changed.
Point 11: Again in Fig 1, the number of isolates (703) is not as reported in the text (983 at line 235) add somewhere in the Figure the correct number of total isolates in each type of sample
Response 11: There was a mistake. The correct numbers were those in the figure and it has been changed in the text. Please see the attachment, line 237.
Point 12: Table 3 change the column Frequency (%) with n (%). Report in the text why your samples are representative of the farms in the Castilla-la Mancha region and how many samples were done for each farm and in total
Response 12: Thanks for your valuable comment. Please see the attachment, Table 3 .

Round 3
Reviewer 2 Report
In my opinion the work, so improved, deserves the publication in Animals.